# Inverse Design of Micro Phononic Beams Incorporating Size Effects via Tandem Neural Network

**DOI:** 10.3390/ma16041518

**Published:** 2023-02-11

**Authors:** Jingru Li, Zhongjian Miao, Sheng Li, Qingfen Ma

**Affiliations:** 1School of Mechanical and Electrical Engineering, Hainan University, Haikou 570228, China; 2State Key Laboratory of Structural Analysis for Industrial Equipment, School of Naval Architecture, Faculty of Vehicle Engineering and Mechanics, Dalian University of Technology, Dalian 116024, China

**Keywords:** inverse design, tandem neural network, size effects, band gaps, microphononic crystal

## Abstract

Phononic crystals of the smaller scale show a promising future in the field of vibration and sound reduction owing to their capability of accurate manipulation of elastic waves arising from size-dependent band gaps. However, manipulating band gaps is still a major challenge for existing design approaches. In order to obtain the microcomposites with desired band gaps, a data drive approach is proposed in this study. A tandem neural network is trained to establish the mapping relation between the flexural wave band gaps and the microphononic beams. The dynamic characteristics of wave motion are described using the modified coupled stress theory, and the transfer matrix method is employed to obtain the band gaps within the size effects. The results show that the proposed network enables feasible generated micro phononic beams and works better than the neural network that outputs design parameters without the help of the forward path. Moreover, even size effects are diminished with increasing unit cell length, the trained model can still generate phononic beams with anticipated band gaps. The present work can definitely pave the way to pursue new breakthroughs in micro phononic crystals and metamaterials research.

## 1. Introduction

Phononic crystals (PnCs) and metamaterials (MMs) have become notable candidates in composite structures and materials owing to their unique wave dispersions and effective characteristics such as wave focusing, wave collimation, wave diffraction and wave absorption and have been widely applied in the wave manipulation field. Among these intriguing functionalities, the band gaps, appearing as a result of the wave motion in such composites, have gained increasing attention for reducing vibration and sound [1,2,3,4] since the elastic and acoustic waves are forbidden from propagating through the frequency regions inside the band gaps. Using different types of mechanisms, Bragg band gaps [5,6], local resonance gaps [7,8,9] and inertial induced gaps [10,11] were investigated to satisfy various structural requirements of applications in vibration and noise control. Numerous studies [12,13,14] have demonstrated that structural configuration and material properties play a significant role in determining wave propagation and attenuation performance regardless of whether underlying mechanisms are used.

In general, topology optimization (TO) has been regarded as a prominent tool to implement the inverse design of the man-made structures for acquiring target wave functionalities. Focusing on enhancing the decaying level of wave band gaps, innovation in structural configurations of PnCs and MMs are discovered using TO within gradient-based or gradient-free algorithms. Li et al. proposed a multiple band gap TO strategy to maximize the specified number of relative band gaps of the in-plane or out-of-plane wave modes for a two-dimensional PnC and the validity of the optimized designs is confirmed by the transmission calculations [15]. Incorporating the genetic algorithm (GA), TO modelling is used for PnC thin plates composed of aluminum and epoxy resin with the objective of broadening the relative bandwidth [16], where the optimal configuration shows diverse features with respect to different filling rate. Ultra-wide omnidirectional band gaps of three-dimensional PnCs were achieved using TO with the fixed-grid finite element method [17]. Additionally, TO is also applied to obtain desirable effective properties that are not available in nature [18,19,20,21] and perform band gap control on metamaterials. Sharma et al. presented the TO framework to achieve the optimal topology of two-phase, soft compressible periodic composites corresponding to the maximum longitudinal wave bandwidth where the nonlinear neo-Hookean material model was employed to characterize the constitutive behavior of soft compressible laminate phases [22]. In order to obtain low-frequency extreme broad band gaps of hierarchical honeycomb metamaterials, the topology of filling scatter was optimized based on the objective function of relative bandwidths of multiple stop bands [23]. The results of the work conducted by Liu et al. indicated that the optimal distribution of the seismic metamaterials using TO share similar topological features; that is, slim connections and large masses, which is helpful in enhancing the local resonance mode [24]. However, the inverse design via the TO approach is updated at each iteration, which makes the numerical simulations recurrent as well as time demanding. On the other hand, the sensitivity analysis essential to gradient-based TO algorithms takes great effort, especially for multifield coupling periodic structures owing to the super nonlinear mapping relation and higher concavity of the design space.

Artificial neural networks (ANNs) have been developed since 1980s, accompanied by the original backpropagation algorithm [25]. The phrase “deep learning” (DL) was first coined by Hinton [26], by which remarkable achievements have been obtained in the fields of computer vision [27], speech recognition [28], decision making [29] and so on, showing a promising future for dealing with the problems of the inverse design of realistic structures and materials, as the underlying nature steps away from the data-driven path. Stimulated by this interactive feature without concurrent numerical simulations, the DL model based on a neural network has been widely applied to study the electromagnetic response for given structures [30,31], the constitutive of solid materials [32], the manipulation of low-frequency acoustic waves [33], the photonic and phononic topological state [34,35], the electric and magnetic dipoles [36] and so on, which further promotes the development of the optimization of PCs and MMs for anticipated band gap properties. Regarding one-dimensional PnCs and elastic MMs, various DL NNs based on the multilayer perceptron (MLP) have been established. By employing the tandem NN, the inverse designs of two-phase PnCs were realized and the results showed that the band gaps of the shear wave of the designed PCs are consistent with the target band gaps [37]. Apart from performing the inverse design directly based on MLPs, Liu et al. regarded NNs as the surrogate model for predicting the mechanical properties of the curved beam metamaterial in their optimization. Similarly, optimal parameters of active metabeams with shunted circuits were acquired quickly for the designed attenuation performance by GA where the machine learning based-model was employed, taking parameters as inputs and the dynamic response as outputs [38]. Regarding two-dimensional PnCs and MMs, convolution neural networks (CNNs) [39] are frequently applied to design the material distribution in-plane, as it they are useful for feature extraction from images and mapping the images to responses. Furthermore, various frameworks of NNs have been proposed by combining the MLP and CNN for accuracy and stability requirements, such as the generative adversarial network (GAN) [40], autoencoder (AE) [41] and other variant forms, which has recently attracted attention of researchers concentrated on tuning wave dispersions. Employing AE and MLP, the topological features of two-dimensional PnCs were extracted and the relationship between the band gaps and topological features was acquired in the study conducted by Li et al. [42]. As a result, PnCs with anticipated band gaps can be generated using the DL model. Jiang et al. established the mapping between the full diagram of real dispersion relation curves and the structural topology of in-plane PnCs via the conditional GAN, based on which the dispersion relation can be proactively tailored within diverse configurations [43]. Focusing on the transmission behavior of acoustic metamaterial, the conditional GAN is applied to generate the cell candidate relating to desired transmission loss of plane waves [44]. It can be noticed that the inverse design of PnCs and MMs using NNs is at the initial stage for target wave propagation and attenuation behavior, which indicates huge research value and tremendous potential to create innovative structures in more a complicated environment or within higher-order elastic theory.

In particular, for the applications at micro- or nanoscale, such as micro architecture lattices, micro actuators, biological micro tubules and nanoelectromechanical systems, wave band gaps possess size-dependent characteristics whose performance cannot be predicted accurately when using classical elasticity theory, and higher-order continuum theories [45,46] need to be applied to PnCs and MMs for capturing the size effects on dispersion relations. Among the nonclassical theories, modified couple stress theory (MCST) [47,48] within the symmetric couple stress tensor has attracted increasing attention, as only one length scale parameter is required to calculate the band structure. Using MCST, the Euler beam with surface energy, and rotation inertia [49], the micro-planar lattice, made up of functionally graded materials [50,51]; the two-dimensional periodic orthotropic composite, containing coated star-shaped inclusions [52]; and the three-dimensional two phase composites at smaller scale [53] were investigated, respectively, to examine the band structure when size effects are taken into account. However, to the best knowledge of authors, the studies aimed at implement the inverse design of periodic composites incorporating size effects are rarely found and need to be explored. Inspired by the facts, and the efficiency of artificial NNs, the current work attempts to generate desirable material properties of the micro-PnC beam using the DL model based on NNs, and the size-dependent band structure of flexural waves is regarded as the interested dynamic response.

Following the introduction, Section 2 provides the derivation of the dispersion relation analysis of a Euler beam with microstructural effects and the description of the DL model based on tandem NNs for acquiring the material configuration of the two-phase micro-PnC beam. Then, the path to predicting size-dependent band gap characteristics and the inverse path for obtaining desirable PnC beams are presented, respectively, in Section 3. The comparisons of the attenuation performance between the results from the generated and real microstructures are also shown to validate the accuracy and efficiency of the established tandem NN model. Lastly, Section 4 provides conclusions.

## 2. Theoretical Formulations and Methods

### 2.1. Model Descriptions of PnC Beam Incorporating Size Effects

Using Bloch theory, a representative cell with less computational costs was modeled to investigate wave motion through the infinite periodic medium. As a consequence, the periodic arrangement of the unit cells comprises the infinite lattice.

As shown in Figure 1, two materials were displayed periodically to compose the PnC beam. The unit cell length is denoted by *L* and the cross-section is depicted by the thickness *h* and width *b*. For linear elastic and isotropic materials, the constitutive relations can be expressed as following, according to MCST:(1)$$\begin{array}{l} \sigma_{ij}=\lambda \varepsilon_{kk}\delta_{ij}+2\mu \varepsilon_{ij}\\ m_{ij}=2l^{2}\mu \chi_{ij} \end{array}$$

In Equation (1), *λ* and *μ* represent the Lamé constants and *l* is responsible for measuring the couple stress effects, usually called the material length scale parameter. *δ_ij_*, *σ_ij_* and *m_ij_* denote the Kronecker delta, the Cauchy stress tensor and the deviatoric part of the couple stress tensor, respectively. The infinitesimal strain and the symmetric curvature tensor are represented by *ε_ij_* and *χ_ij_*, the derivations of which are
(2)$$\begin{array}{l} \varepsilon_{ij}=\frac{1}{2}\left(u_{i,j}+u_{j,i}\right)\\ \chi_{ij}=\frac{1}{2}\left(\theta_{i,j}+\theta_{j,i}\right) \end{array}$$

Here, *u_i_* represents the displacement field component and *θ_i_* denotes the component of the rotation vector, which can be acquired with
(3)$$\theta_{i}=\frac{1}{2}\varepsilon_{ijk}u_{k,j}$$

Following the Euler beam theory, the nonzero components of **σ**, **ε**, **m** and **χ** can be expressed as
(4)$$\left\{ \begin{array}{l} \varepsilon_{xx}=-z\frac{\partial^{2}w}{\partial x^{2}}\\ \sigma_{xx}=-zE\frac{\partial^{2}w}{\partial x^{2}} \end{array}\right.,\;\;\;\left\{ \begin{array}{l} \chi_{xy}=-\frac{1}{2}\frac{\partial^{2}w}{\partial x^{2}}\\ m_{xy}=-\mu l_{0}^{2}\frac{\partial^{2}w}{\partial x^{2}} \end{array}\right.$$

The total internal energy and kinetic energy can be obtained with
(5a)$$U=\frac{1}{2}\left\{\underset{V_{b}}{∭}\left(\sigma_{xx}\varepsilon_{xx}\right)dV_{b}+\underset{V_{b}}{∭}\left(m_{xy}\chi_{xy}\right)dV_{b}\right\}$$
(5b)$$T=\int_{0}^{L}\rho A{\dot{w}}^{2}dx$$
where *V_b_* and *A* represent the volume and the area of the cross-section of the unit cell. The overhead “·” denotes the first time derivative. Substituting Equation (4) with Equation (5a,b) and then performing variational operations on the potential and kinetic energy as well as applying Hamilton’s principle, the dynamic equation for describing the motion of the PnC beam with scaling effects can be obtained:(6)$$\left(EI+GAl_{0}^{2}\right)\frac{\partial^{4}w}{\partial x^{4}}+\rho A\frac{\partial^{2}w}{\partial t^{2}}=q\left(x\right)$$
where *I* and *A* are the second moment of inertia and the area of the cross section.

### 2.2. Dispersion Analysis using the Transfer Matrix Method

Looking at Equation (6), the displacement field for the *n*^th^ unit cell can be described as follows:(7)$$w_{I}^{n}=e^{j\omega t}\sum_{m=1}^{4}W_{m}^{n}e^{-jk_{m}^{I}x},\;\;\;w_{{}_{II}}^{n}=e^{j\omega t}\sum_{m=1}^{4}U_{m}^{n}e^{-jk_{m}^{II}x}$$

Superscript I and II denote the constituent made up of material I and material II, respectively, and the characteristic wavenumber *k* is acquired by solving Equation (6). Simultaneously, the continuity conditions at the interface between two materials require that the displacement, rotation angle, bending moment and shear force be
(8)$$\begin{array}{l} w_{I}^{n}\left(L/2\right)=w_{II}^{n}\left(L/2\right)\\ \frac{\partial w_{I}^{n}}{\partial x}\left(L/2\right)=\frac{\partial w_{II}^{n}}{\partial x}\left(L/2\right)\\ D_{I}\frac{\partial^{2}w_{I}^{n}}{\partial x^{2}}\left(L/2\right)=D_{II}\frac{\partial^{2}w_{II}^{n}}{\partial x^{2}}\left(L/2\right)\\ D_{I}\frac{\partial^{3}w_{I}^{n}}{\partial x^{3}}\left(L/2\right)=D_{II}\frac{\partial^{3}w_{II}^{n}}{\partial x^{3}}\left(L/2\right) \end{array}$$
where *D*_I_ = *E*_I_*I*_I_ + *GA*(*l*_I_)^2^; *D*_II_ = *E*_II_*I*_II_ + *GA*(*l*_II_)^2^. Similarly, for the contact face between *n*−1^th^ and *n*^th^ unit cells, the boundary conditions should follow
(9)$$\begin{array}{l} w_{II}^{n-1}\left(L\right)=w_{I}^{n}\left(0\right)\\ \frac{\partial w_{II}^{n-1}}{\partial x}\left(L\right)=\frac{\partial w_{I}^{n}}{\partial x}\left(0\right)\\ D_{II}\frac{\partial^{2}w_{II}^{n-1}}{\partial x^{2}}\left(L\right)=D_{I}\frac{\partial^{2}w_{I}^{n}}{\partial x^{2}}\left(0\right)\\ D_{II}\frac{\partial^{3}w_{II}^{n-1}}{\partial x^{3}}\left(L\right)=D_{I}\frac{\partial^{3}w_{I}^{n}}{\partial x^{3}}\left(0\right) \end{array}$$

Substituting Equation (7) with Equations (8) and (9) and then applying the transfer matrix method yields the results of transmissibility between the displacement coefficient vectors:(10)$$\begin{array}{c} Td_{II}^{n-1}=d_{II}^{n}\\ T={\left(H_{2}\right)}^{-1}H_{1}{\left(H_{4}\right)}^{-1}H_{3} \end{array}$$
where **T** denotes the transfer matrix for the micro PnC beam. The expressions of **H**_1_, **H**_2_, **H**_3_ and **H**_4_ are:(11a)$$H_{1}=\left[\begin{array}{l} e^{jk_{1}^{I}L/2}\;\;\;\;\;\;\;\;\;\;\;\;\;\;e^{jk_{2}^{I}L/2}\;\;\;\;\;\;\;\;\;\;\;\;\;\;\;e^{jk_{3}^{I}L/2}\;\;\;\;\;\;\;\;\;\;\;\;\;\;\;\;e^{jk_{4}^{I}L/2}\\ jk_{1}^{I}e^{jk_{1}^{I}L/2}\;\;\;\;\;\;\;\;\;\;jk_{2}^{I}e^{jk_{2}^{I}L/2}\;\;\;\;\;\;\;\;\;\;jk_{3}^{I}e^{jk_{3}^{I}L/2}\;\;\;\;\;\;\;\;\;\;\;jk_{4}^{I}e^{jk_{4}^{I}L/2}\\ -D_{I}{\left(k_{1}^{I}\right)}^{2}e^{jk_{1}^{I}L/2}\;-D_{I}{\left(k_{2}^{I}\right)}^{2}e^{jk_{2}^{I}L/2}\;-D_{I}{\left(k_{3}^{I}\right)}^{2}e^{jk_{3}^{I}L/2}\;-D_{I}{\left(k_{4}^{I}\right)}^{2}e^{jk_{4}^{I}L/2}\\ -jD_{I}{\left(k_{1}^{I}\right)}^{3}e^{jk_{1}^{I}L/2}\;-jD_{I}{\left(k_{2}^{I}\right)}^{3}e^{jk_{2}^{I}L/2}\;-jD_{I}{\left(k_{3}^{I}\right)}^{3}e^{jk_{4}^{I}L/2}\;-jD_{I}{\left(k_{4}^{I}\right)}^{3}e^{jk_{4}^{I}L/2} \end{array}\right]$$
(11b)$$H_{2}=\left[\begin{array}{l} e^{jk_{1}^{II}L/2}\;\;\;\;\;\;\;\;\;\;\;\;\;\;e^{jk_{2}^{II}L/2}\;\;\;\;\;\;\;\;\;\;\;\;\;\;\;e^{jk_{3}^{II}L/2}\;\;\;\;\;\;\;\;\;\;\;\;\;\;\;\;e^{jk_{4}^{II}L/2}\\ jk_{1}^{II}e^{jk_{1}^{II}L/2}\;\;\;\;\;\;\;\;\;\;jk_{2}^{II}e^{jk_{2}^{II}L/2}\;\;\;\;\;\;\;\;\;\;jk_{3}^{II}e^{jk_{3}^{II}L/2}\;\;\;\;\;\;\;\;\;\;\;jk_{4}^{II}e^{jk_{4}^{II}L/2}\\ -D_{II}{\left(k_{1}^{II}\right)}^{2}e^{jk_{1}^{II}L/2}\;-D_{II}{\left(k_{2}^{II}\right)}^{2}e^{jk_{2}^{II}L/2}\;-D_{II}{\left(k_{3}^{II}\right)}^{2}e^{jk_{3}^{II}L/2}\;-D_{II}{\left(k_{4}^{II}\right)}^{2}e^{jk_{4}^{II}L/2}\\ -jD_{II}{\left(k_{1}^{II}\right)}^{3}e^{jk_{1}^{II}L/2}\;-jD_{II}{\left(k_{2}^{II}\right)}^{3}e^{jk_{2}^{II}L/2}\;-jD_{II}{\left(k_{3}^{II}\right)}^{3}e^{jk_{4}^{II}L/2}\;-jD_{II}{\left(k_{4}^{II}\right)}^{3}e^{jk_{4}^{II}L/2} \end{array}\right]$$
(11c)$$H_{3}=\left[\begin{array}{l} e^{jk_{1}^{II}L}\;\;\;\;\;\;\;\;\;\;\;\;\;\;e^{jk_{2}^{II}L}\;\;\;\;\;\;\;\;\;\;\;\;\;\;\;e^{jk_{3}^{II}L}\;\;\;\;\;\;\;\;\;\;\;\;\;\;\;\;e^{jk_{4}^{II}L}\\ jk_{1}^{II}e^{jk_{1}^{II}L}\;\;\;\;\;\;\;\;\;\;jk_{2}^{II}e^{jk_{2}^{II}L}\;\;\;\;\;\;\;\;\;\;jk_{3}^{II}e^{jk_{3}^{II}L}\;\;\;\;\;\;\;\;\;\;\;jk_{4}^{II}e^{jk_{4}^{II}L}\\ -D_{II}{\left(k_{1}^{II}\right)}^{2}e^{jk_{1}^{II}L}\;-D_{II}{\left(k_{2}^{II}\right)}^{2}e^{jk_{2}^{II}L}\;-D_{II}{\left(k_{3}^{II}\right)}^{2}e^{jk_{3}^{II}L}\;-D_{II}{\left(k_{4}^{II}\right)}^{2}e^{jk_{4}^{II}L}\\ -jD_{II}{\left(k_{1}^{II}\right)}^{3}e^{jk_{1}^{II}L}\;-jD_{II}{\left(k_{2}^{II}\right)}^{3}e^{jk_{2}^{II}L}\;-jD_{II}{\left(k_{3}^{II}\right)}^{3}e^{jk_{4}^{II}L}\;-jD_{II}{\left(k_{4}^{II}\right)}^{3}e^{jk_{4}^{II}L} \end{array}\right]$$
(11d)$$H_{4}=\left[\begin{array}{l} 1\;\;\;\;\;\;\;\;\;\;\;\;\;\;1\;\;\;\;\;\;\;\;\;\;\;\;\;\;\;1\;\;\;\;\;\;\;\;\;\;\;\;\;\;\;1\\ jk_{1}^{I}\;\;\;\;\;\;\;\;\;\;jk_{2}^{I}\;\;\;\;\;\;\;\;\;\;jk_{3}^{I}\;\;\;\;\;\;\;\;\;\;\;jk_{4}^{I}\\ -D_{I}{\left(k_{1}^{I}\right)}^{2}\;-D_{I}{\left(k_{2}^{I}\right)}^{2}\;-D_{I}{\left(k_{3}^{I}\right)}^{2}\;-D_{I}{\left(k_{4}^{I}\right)}^{2}\\ -jD_{I}{\left(k_{1}^{I}\right)}^{3}\;-jD_{I}{\left(k_{2}^{I}\right)}^{3}\;-jD_{I}{\left(k_{3}^{I}\right)}^{3}\;-jD_{I}{\left(k_{4}^{I}\right)}^{3} \end{array}\right]$$

According to Bloch theory, the periodic condition ought to be applied:(12)$$d_{II}^{n}=\exp(-jqL)d_{II}^{n-1}$$

Here, *q* is the Bloch wavenumber. Substituting Equation (10) with Equation (12) leads to:(13)|T−exp(−jqL)I|=0

The real part and imaginary part of *q* are obtained by solving Equation (13), which regards *q* as the eigenvalues for the given frequencies, in which case the location, width and attenuation level of flexural wave band gaps can be examined. Using the numerical model, the band gaps with size effects are can be predicted within the arbitrary design of the micro-PnC beam.

### 2.3. The Neural Network Model

In this section, the basic theories of the artificial neural network and tandem neural network used for the inverse design of the micro-PnC beam are introduced.

#### 2.3.1. Basis of Artificial Neural Networks

In order to acquire a feasible design for the micro-PnC beam—the typical one-dimensional periodic lattice—the MLP is adopted for mapping the target band gaps incorporating size effects into the interested design variables. In the present work, the normalized bounding frequencies of the first and second stop bands are applied to reflect the flexural wave band gap behavior and the aim is to generate accurate, size-dependent material parameters for anticipated band gap characteristics.

It is acknowledged that the MLP generally consists of an input layer, output layer and multiple hidden layers where each neuron in one layer is connected to all neurons of next layer in the absence of interlayer connections. The input data are processed layer after layer using the weights (*w*), bias (*b*) and activate functions *σ*(•) in order to match the output targets. In general, the weights and bias connecting the *i*^th^ neuron in the (*n*−1)^th^ layer to the *j*^th^ neuron in the *n*^th^ layer are denoted by $w_{ji}^{m}$ and $b_{j}^{n}$ and the output for the *j*^th^ neuron in the *n*^th^ layer $a_{j}^{n}$ can be calculated via the weighted summation of the outputs from the previous layer with the help of nonlinear activation functions:(14)$$a_{j}^{n}=\sigma \left(\sum_{i}w_{ji}^{n}a_{i}^{n-1}+b_{j}^{n}\right)=\sigma \left(z_{j}^{n}\right)$$

Furthermore, a cost function C(•) is defined to evaluate network performance, the error of which is minimized by adjusting the weights and bias of the entire layers of the neural network (NN). As the design of the micro-PnC beam is indicated by a continuous vector, the mean squared error (MSE) is applied as the cost function in this work:(15)$$C=\frac{1}{mn}\sum_{i=1}^{m}\sum_{j=1}^{n}{\left(y_{ij}-{\tilde{y}}_{ij}\right)}^{2}$$
where *m* and *n* represent the number of input datasets and the dimension of the output. *y_ij_* is the *j*^th^ ground truth feature of the *i*^th^ set of data and ${\tilde{y}}_{ij}$ is the output of the corresponding feature in the *i*^th^ dataset. The process of minimizing *C* is equivalent to training NNs, which is accomplished using the backpropagation method to calculate the gradient for updating the weights and bias vectors.

The Adam algorithm proved efficient in previous works [54,55] and is attracting increasing attention for optimizing NNs. The flowing equations show the flowchart for updating **θ** at the *i*^th^ iteration:(16a)$$\beta g_{i}=\nabla_{\theta }C_{i}\left(\theta_{i-1}\right)$$
(16b)$$m_{i}=\beta_{1}m_{i-1}+\left(1-\beta_{1}\right)g_{i}$$
(16c)$$v_{i}=\beta_{2}m_{i-1}+(1-\beta_{2})g_{i}^{2}$$
(16d)$${\tilde{m}}_{i}=m_{i}/\left(1-{\left(\beta_{1}\right)}^{i}\right)$$
(16e)$${\tilde{v}}_{i}=v_{i}/\left(1-{\left(\beta_{2}\right)}^{i}\right)$$
(16f)$$\theta_{i}=\theta_{i-1}-\alpha {\tilde{m}}_{i}/\left(\sqrt{{\tilde{v}}_{i}}+\varepsilon \right)$$

Here *β*_1_ and *β*_2_ are the two hyperparameters that should be provided in advance in the Adam algorithm and are assumed as 0.9 and 0.999, respectively. *α* denotes the learning rate that reduces gradually after a certain amount of training epochs in this work, whose initial value is designated as 0.005. The biased first moment and second raw moment estimates are denoted by **m***_i_* and **v***_i_* at the *i*^th^ iteration.

#### 2.3.2. The Tandem Framework for Inverse Design

In order to avoid the uncertainty, caused by the nature of real structures, that various material assemblies can result in the same dynamic response, a tandem architecture that exhibits a bidirectional configuration is applied, which is able to allow data to flow in both a forward path and an inverse path.

As shown in Figure 2, the forward network learns the mapping relation by using a supervised learning method where the inputs are design parameters, and the labels are corresponding band gap characteristics calculated using the TMM model. In the tandem NN, the forward network is served as the pretrained network. It is noted that length scaling effects are taken into account in band gap analysis. The activation functions that take part in the forward path NN are tanh- and ReLU functions whose expressions are
(17)$$\begin{array}{l} \sigma_{tanh}=\frac{e^{x}-e^{-x}}{e^{x}-e^{-x}}\\ \sigma_{relu}=\max\left(0,\;x\right) \end{array}$$

It can be seen that the tandem NN is trained in advance and the inverse network is trained in the full loop with the help of a well-trained forward path. Obviously, the inverse path is processed in an unsupervised form, which regards the target band gap properties as inputs and the outputs of which become the inputs of the pretrained NN model in the forward path. The goal of all the NNs is to generate feasible designs based on which the calculated multiple stop bands agree well with those of the real microstructures. Additionally, the errors from the forward path and inverse path are regarded as the cost function and require minimization to enhance the robustness of the NNs.

After training the tandem NN, the design of micro-PnC beams can be output with respect to assigned bounding frequencies of the first and second stop bands incorporating size effects. Figure 3 illustrates the NN model as the analytical model that replaces the optimization model, completing the inverse design of the PnC beam at the smaller scale.

## 3. Results and Discussions

The geometrical parameters of the micro-PnC beam remain constant during the inverse design process, and detailed information can be found in Table 1. To quantitatively illustrate the generated structures at the smaller scale, the flexural wave band structure of the phononic microbeam is obtained using MCST and classical elasticity theory within TMM models, respectively, then taken as the labels of the deep learning model in the forward path, under which each constructed network is trained separately. Noting that the Poisson’s ratio is excluded in the TMM model, the normalized frequency defined as following is adopted to identify the dispersion relation from low to high locations:(18)$$\Omega =fL^{2}\sqrt{\rho_{I}A_{I}}/\sqrt{E_{I}I_{I}}$$
where the material properties of material I is described using steel; that is, *E*_I_ = 177.3 GPa, *ρ*_I_ = 7000 kg/m^3^ and *l*_I_ = 6.76 μm.

Concerning the flexural wave propagation and attenuation—the ratios *E_r_*, *ρ_r_* and *l_r_* between the elastic modulus, the density and the length scale parameter are considered as main features to adjust the band structure; meanwhile, the former two are regarded as the physical parameters used in the forward and inverse design when applying the classical elasticity model. Table 2 provides the selected values of these design parameters.

### 3.1. Forward Design of Micro-PnC Beams

As seen from Table 2, there are 12^3^ sets of the data whose target band gap characteristics are predicted via the current nonclassical model and classical model. Seventy-five percent of the datasets are randomly selected to train the NNs, 25% of which serve as the test set. When focusing on the lower and upper bounding frequencies Ω*_iL_* and Ω*_iU_* of the *i*-th stop band of the nonclassical model, the architecture of the NN consists of five fully connected layers, each of which containing 100, 60, 40, 10 and 4 neurons. In order to add nonlinearity, the input layer is associated with the tanh-function and the ReLU functions are applied in the hidden layers. If the classical elasticity theory is adopted, the neurons in each layer are modified to 80, 60, 30,10 and 4. The cost function is formulated in mean square error form.

After training the NNs, Figure 4a,b presents the correlation of bounding frequencies predicted by the forward NN and TMM models with and without size effects. As can be observed, the predicted values match the ground truth well whether the size-dependent behavior is taken into consideration or not. This performance demonstrates that the network is a powerful and efficient tool for acquiring band gap characteristics of small-scale flexural motion. Specifically, the band gap properties predicted by the NNs are shown in Figure 4c,d between four random examples, noting that the four designs are produced randomly and not included in the training and test datasets. Moreover, Table 3 provides the values of these input features. From the comparison between NN predictions and numerical simulations, one can conclude that the network can accurately examine the band gap’s properties. Moreover, as seen from the scale of the normalized frequency shown in Figure 4c,d, the addition of size effects is able to lift the band structure.

### 3.2. Forward Network with Different Unit Cell Lengths

Accompanying the same architecture of the forward network and the remaining training dataset, the band gaps are predicted by the NNs with different unit cell lengths *L*, in which case, the length *L* is assumed as 0.0005 m, 0.002 m and 0.01 m, respectively. The corresponding results for the random examples are shown from Figure 5, Figure 6, Figure 7 and Figure 8, where the predictions and simulations agree with each other well with respect to different unit cell lengths.

Obviously, with the increase of *L*, the size-dependent Bragg band gaps move towards lower frequencies and the NN model can reflect the trend accurately. In contrast, as observed from the NN and TMM models without size effects, the location of the stop bands depicted by normalized frequencies is insensitive to the unit cell length. In addition, the differences caused by the scaling effects are diminished as the unit cell length increases. From the following plots, one can notice that these values are merely overlapped when *L* is modified to 0.01 m.

### 3.3. Inverse Design of Micro-PnC Beams

In this subsection, the inverse design in a direct way is considered at first. In this case, the supervised network is applied, which regards the target bounding frequencies as inputs and material parameters as outputs. Noting that the DL models for generating material parameters containing the length scale parameter or not share same architecture except for the output layer, which is composed of three neurons in the former case and two neurons in the latter case. Using MCST and classical elasticity theory, Figure 9 and Figure 10 show the correlations of the outputs from the NNs and the true parameters as well as the bounding frequencies calculated based on these parameters.

The mean relative errors of the bounding frequencies MRE are also provided. Comparing Figure 9 and Figure 10, it can be seen that higher accuracy is obtained between the results of flexural wave band gaps with the classical theory from the inverse design for micro-PnC beams. This is because the inverse NNs inputting band gaps from MSCT modelling adds the length scale parameter as output, the dependency of which on the band gaps differs significantly compared to the modulus and the density. As a result, the dispersion curves with negligible differences can be produced using different assembly of design parameters and the phenomenon is further deteriorated when the size-dependent behavior is considered.

In order to avoid the inaccuracy of the inverse design caused by the “one to many” phenomenon, a tandem network is established, the training process is implemented in an unsupervised way and the pretrained forward network is connected to the inverse path. The structure of the NN is 4–100–50–30–10–3–100–60–40–10–4 and the function of the intermediate layer containing three neurons is to combine the two paths, the output of which essentially denotes the predicted design of the PnC beam. It is worth noting that the weights and bias vectors of the pretrained forward network remain constant during the training process. Regarding the former part, the tanh- and ReLU functions are applied at the input layer and hidden layers, respectively, and the final loss function is defined as:(19)$$N_{loss}=Mean\left(\sum {\left(\Omega_{p}-\Omega_{r}\right)}^{2}\right)+\lambda \cdot Mean\left(\sum {\left(x_{p}-x_{r}\right)}^{2}\right)$$
where *λ* = 1 in this work. Figure 11 shows the correlation between the band gap properties within size effects examined by the numerical models based on the generated and real microstructures. As observed from Figure 11, the predictions of the bounding frequencies match well with the ground band gaps and the mean relative errors drop rapidly compared to Figure 9, which validates the efficiency of the tandem framework in improving the accuracy of generating micro-PnC beams for anticipated flexural wave band gaps.

Furthermore, focusing on the four random examples mentioned before, the design parameters of these random cases are predicted as 0.0451, 0.3294, 0.9835; 0.056, 0.4062, 0.9851; 0.0674, 0.2023, 0.9857; and 0.0881, 0.4332, 1.0224; respectively; using the established tandem network. Applying these predictions, the numerical simulations for calculating the complex dispersion curves are performed, from which both propagating and evanescent wave modes can be observed. Moreover, the results are compared to the ground truth in Figure 12.

From Figure 12 and Table 3, it can be found that even the generated PnC beam deviates from the real structure; perfect agreement is achieved between the real and imaginary panels of the complex band structures. Thus, it can be concluded that the established deep learning model can not only accurately produce feasible structures but also retain the diversity of micro PnC beams for desired dispersion relations incorporating size-dependent behavior.

### 3.4. Inverse Design with Different Unit Cell Lengths

Adopting the NN without the well-trained forward path, the relative mean errors between the bounding frequencies are 2.12%, 1.94% and 1.57% for *L* = 0.0005 m, *L* = 0.002 m and *L* = 0.01 m, respectively. As the unit cell length increases, the accuracy improves, which can be explained by the fact that the increasing length decreases the dependency of the length scale parameter on adjusting the band gaps. Once applying the tandem network, the relative mean errors are calculated as 0.37%, 0.21% and 0.19%, definitely enhancing the accuracy of the inverse design with respect to different unit cell lengths.

Then, looking at the random examples accompanying specific unit cell length, when *L* is 0.0005 m, the predicted design parameters are 0.457, 0.3327, 0.9879; 0.0501, 0.4055, 0.9841; 0.0569, 0.2014, 0.9973; and 0.0897, 0.4395, 1.0496; respectively. Applying these generated parameters, the attenuation performance confined to |Imag(kL)|/(2π)≤2 are examined and shown in Figure 13. As seen from Figure 13, the decaying level inside the band gaps are in accordance with the ground truth even for the higher-order evanescent wave modes.

Similarly, the generated parameters of the four random cases are 0.0447, 0.3286, 0.9966; 0.0588, 0.407, 0.9946; 0.071, 0.1995, 0.9986; 0.0856, 0.439, 0.9931; and 0.0449,0.325, 0.9973; 0.0593, 0.3987, 0.998; 0.075, 0.2003, 1.0046; 0.0818, 0.4346, 0.999; where *L* is assumed as 0.002 m and 0.01 m. For these two cases, the generated parameters exhibit slight differences since the size effects gradually become invalid with larger *L*. Still, the corresponding attenuation performance of flexural waves remain consistent at the smaller scale, which can be found from Figure 14 to Figure 15.

## 4. Conclusions

In the present work, the DL model based on the tandem network was applied to generate micro-PnC beams incorporating size effects where the design parameters including the length scale parameter were taken as the outputs and the interested band gap properties described by the lower and upper bounding frequencies of the first and second stop bands were regarded as the inputs. The NN model was established in the MLP form and the forward path was first trained and then transported to the tandem network playing the role of the pretrained NN model to predict the bounding frequencies for a given output from the inverse path. The accuracy of the NN models was assured by the boundary predictions of stop bands between the generated and real microperiodic structures from the test dataset. For comparison purposes, the NN models used to realize the intelligent design of the flexural wave dispersions without size effects were also provided.

The results of design parameters predicted by the inverse models that ignored the forward information indicated that the NN model focused on the design of the PnC beam without size effects worked much better than the NN model concentrating on the generation of PnC beams with size effects. This is because the addition of the length scale parameter induces high inconsistent sensitivity on the band structure, which further aggravated the non-uniqueness problem. The tandem NN avoided this dilemma as the objective was transformed to match the target band gaps and the relative errors of the bounding frequencies were, consequently, significantly reduced. When the numerical cases of different unit cell lengths were taken into account, the tandem NN model could still provide desirable micro-PnC beam designs and perfect agreements were found between the results of the bounding frequencies, the real dispersion curves and the attenuation diagrams from the generated and real microstructures even when the generated candidates deviated from the real designs. In addition, it was found that the differences between the dispersion curves caused by size effects vanished as the unit cell length increased. Furthermore, four random examples selected from the training and test datasets were used and the NN model was able to generate microstructures whose attenuation performance of flexural motion was in accordance with the ground truth, which demonstrates that the NN model based on the data driven approach is a powerful complement to traditional inverse design for innovation structures.

The proposed framework can be extended to map material and structural configurations to the wave motion of tunable MMs at smaller scale based on multifield coupling mechanisms, such as piezoelectric MMs, electromagnetic MMs and fluid-structure interaction MMs, and this work is thus helpful in developing inverse design methods for diverse functionality of wave manipulations in small-scale composites.

## Figures and Tables

**Figure 1 materials-16-01518-f001:**
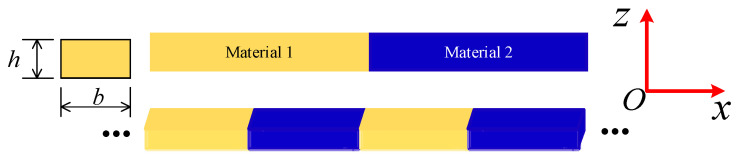
Schematic diagrams of the unit cell model and the infinite lattice of the micro-PnC beam.

**Figure 2 materials-16-01518-f002:**
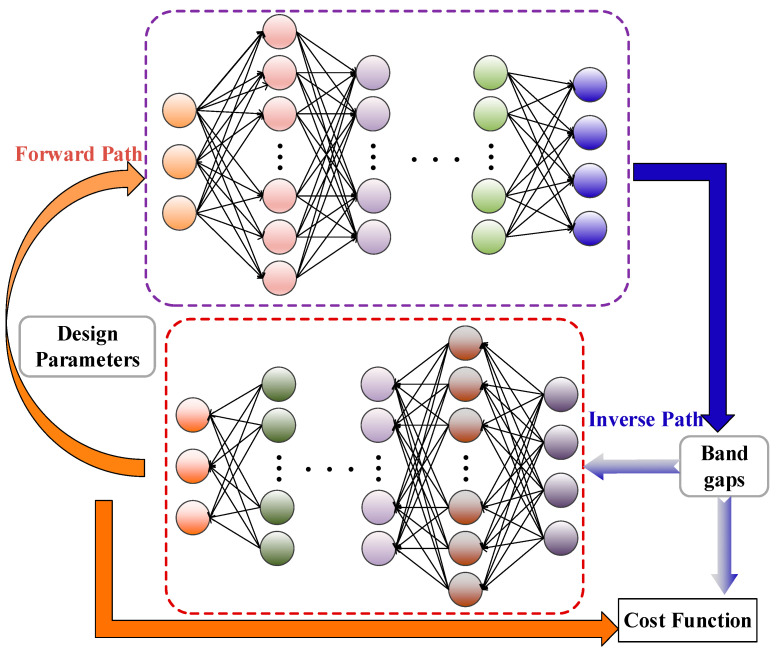
The architecture of the NN model for designing the micro-PnC beam.

**Figure 3 materials-16-01518-f003:**
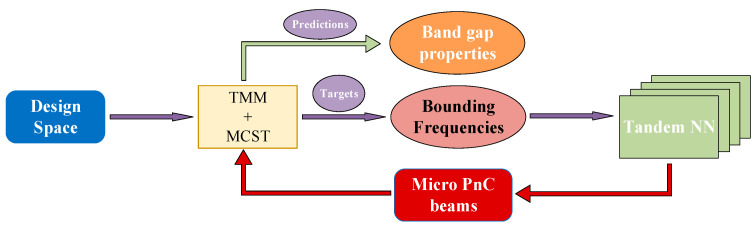
The implement of the inverse design using the MCST and NN models.

**Figure 4 materials-16-01518-f004:**
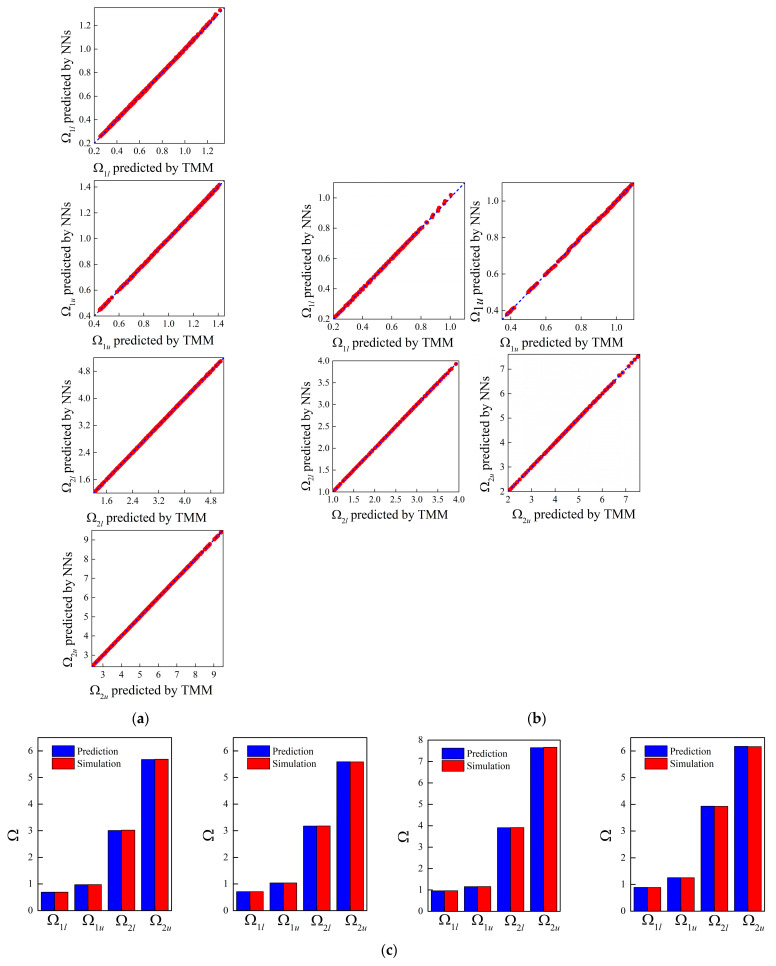
Correlation between the bounding frequencies obtained by the NN and TMM models for the forward design of micro-PnC beams (**a**) incorporating size effects and (**b**) without size effects; (**c**) random examples of band gap characteristics predicted by NNs considering the length scale parameter and (**d**) NNs excluding the length scale parameter. The red column denotes the ground truth of band gaps.

**Figure 5 materials-16-01518-f005:**
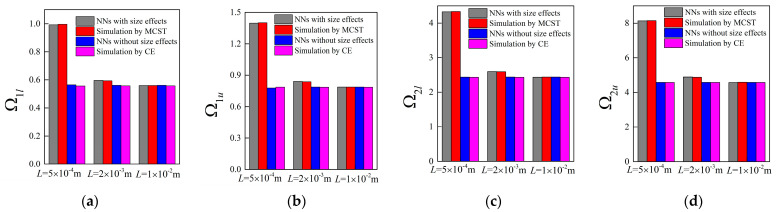
The predictions for bounding frequencies of size-dependent and size-independent band gaps of random example 1: (**a**,**b**) the lower and upper boundary of the first band gap; (**c**,**d**) the lower and upper boundary of the second band gap.

**Figure 6 materials-16-01518-f006:**
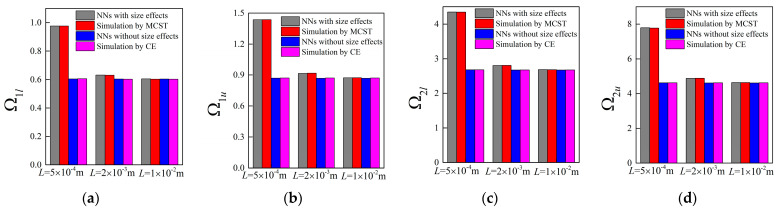
The predictions for bounding frequencies of size-dependent and size-independent band gaps of random example 2: (**a**,**b**) the lower and upper boundary of the first band gap; (**c**,**d**) the lower and upper boundary of the second band gap.

**Figure 7 materials-16-01518-f007:**
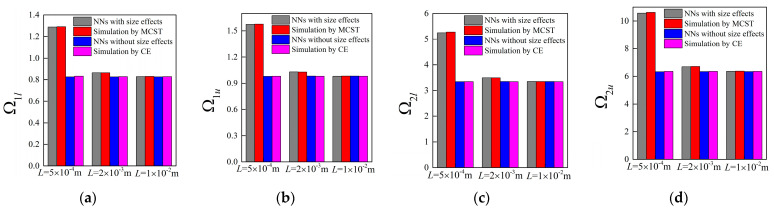
The predictions for bounding frequencies of size-dependent and size-independent band gaps of random example 3: (**a**,**b**) the lower and upper boundary of the first band gap; (**c**,**d**) the lower and upper boundary of the second band gap.

**Figure 8 materials-16-01518-f008:**
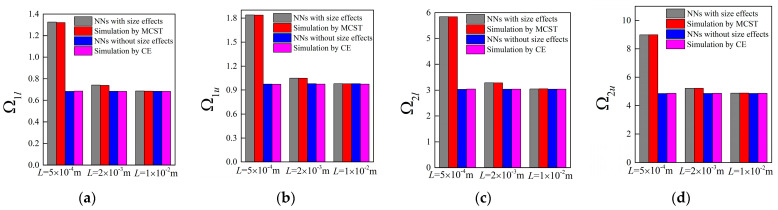
The predictions for bounding frequencies of size-dependent and size-independent band gaps of random example 4: (**a**,**b**) the lower and upper boundary of first band gap; (**c**,**d**) the lower and upper boundary of the second band gap.

**Figure 9 materials-16-01518-f009:**
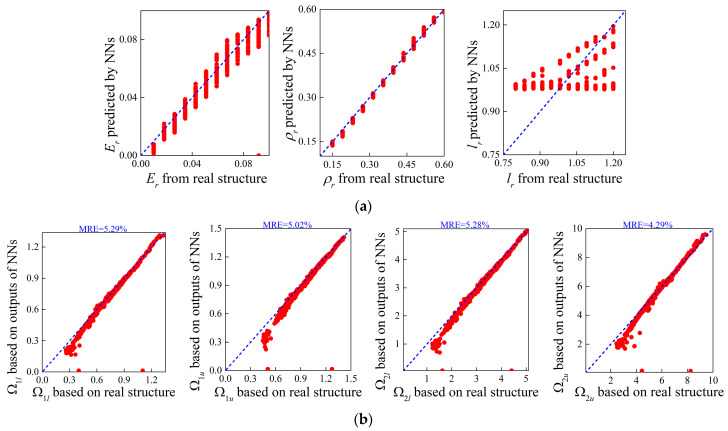
Correlations between (**a**) the outputs of NNs and real design parameters as well as (**b**) the size-dependent band gaps based on these parameters.

**Figure 10 materials-16-01518-f010:**
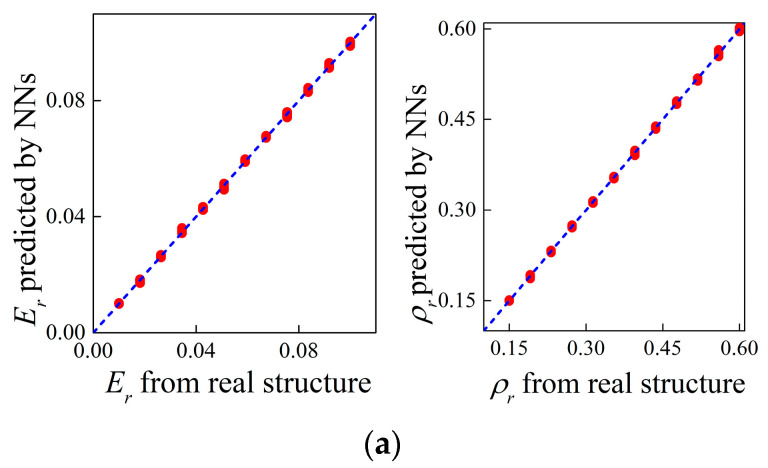
Correlations between (**a**) the outputs of NNs and real design parameters as well as (**b**) the size-independent band gaps based on these parameters.

**Figure 11 materials-16-01518-f011:**
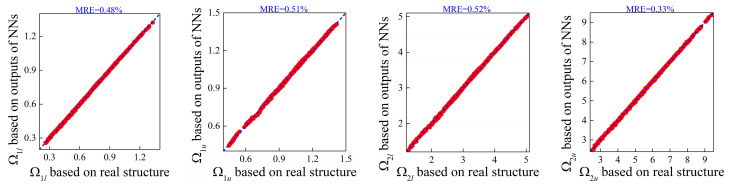
Correlations between the target size-dependent bounding frequencies and those calculated based on the designs output from tandem NNs.

**Figure 12 materials-16-01518-f012:**
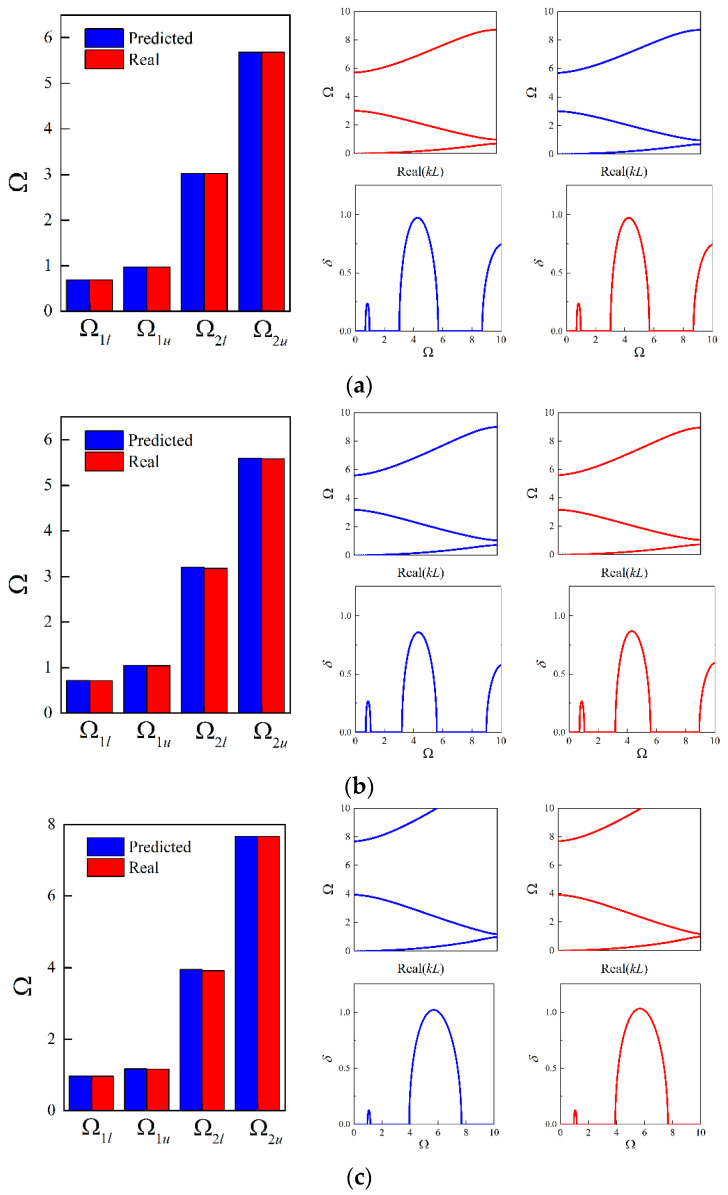
Comparison results of the bounding frequencies and complex dispersion curves between the generated parameters and real parameters for random examples: (**a**) example 1; (**b**) example 2; (**c**) example 3; (**d**) example 4. The blue lines denote the results based on generated values and the red lines denote that of ground truth.

**Figure 13 materials-16-01518-f013:**
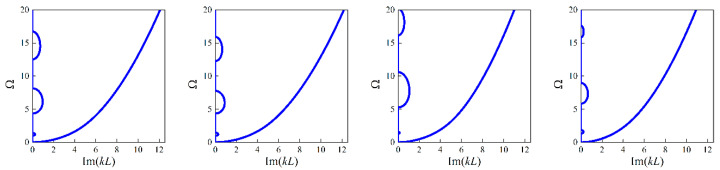
The imaginary panels of flexural waves of the random samples based on the generated structures and real structures with *L* = 0.0005 m: (**a**) example 1; (**b**) example 2; (**c**) example 3; (**d**) example 4. The blue lines denote the results based on generated values and the red lines denote that of ground truth.

**Figure 14 materials-16-01518-f014:**
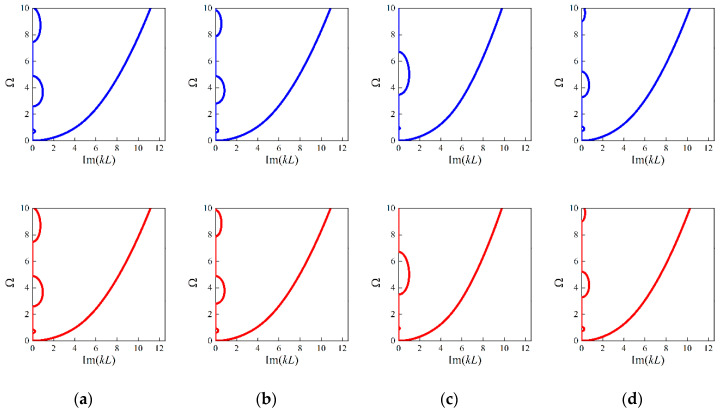
The imaginary panels of flexural waves of the random samples based on the generated structures and real structures with *L* = 0.002 m: (**a**) example 1; (**b**) example 2; (**c**) example 3; (**d**) example 4. The blue lines denote the results based on generated values and the red lines denote that of ground truth.

**Figure 15 materials-16-01518-f015:**
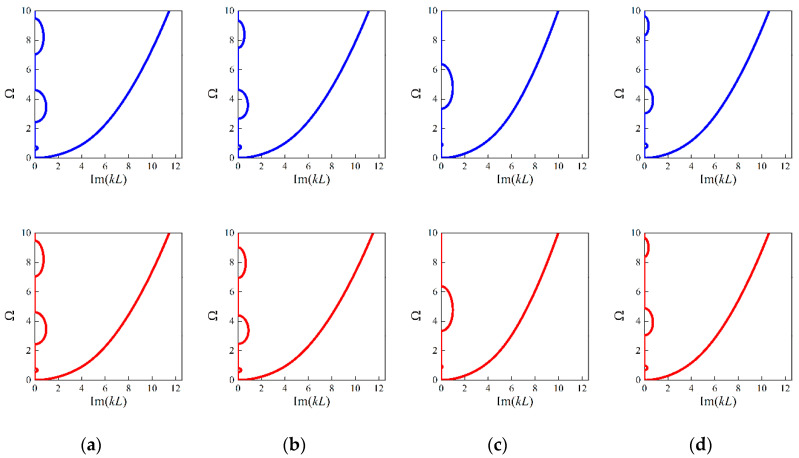
The imaginary panels of flexural waves of the random samples based on the generated structures and real structures with *L* = 0.01 m: (**a**) example 1; (**b**) example 2; (**c**) example 3; (**d**) example 4. The blue lines denote the results based on generated values and the red lines denote that of ground truth.

**Table 1 materials-16-01518-t001:** Geometrical parameters of the micro-PnC beam.

Parameters	Descriptions	Value
*L* (mm)	The unit cell length	1
*h* (mm)	The thickness of microbeam	*L*/50
*b* (mm)	The width of the rectangular section	2*h*
*V_f_*	The volume fraction of material 1	0.5

**Table 2 materials-16-01518-t002:** The selected values of material parameters to compose the dataset.

*E_r_*	*ρ_r_*	*l_r_*
0.0100	0.1500	0.8000
0.0182	0.1909	0.8364
0.0264	0.2318	0.8727
0.0345	0.2727	0.9091
0.0427	0.3136	0.9455
0.0509	0.3545	0.9818
0.0591	0.3955	1.0182
0.0673	0.4364	1.0545
0.0755	0.4773	1.0909
0.0836	0.5182	1.1273
0.0918	0.5591	1.1636
0.1000	0.6000	1.2000

**Table 3 materials-16-01518-t003:** Random designs of material properties for the micro-PnC beam.

Material Features	Example 1	Example 2	Example 3	Example 4
*E_r_*	0.045	0.06	0.75	0.0822
*ρ_r_*	0.33	0.4	0.2	0.433
*l_r_*	0.99	0.85	0.8	1.11

## Data Availability

Not applicable.

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
