# Peer review of "Inverse Design of Micro Phononic Beams Incorporating Size Effects via Tandem Neural Network"

_materials, 2023, doi:10.3390/ma16041518_

Round 1

Reviewer 1 Report

Deep Learning is a broad term, and using it in the title may not lure a potential reader into noticing. I would recommend using your model (algorithm) name in the title.

Check that the abstract provides an accurate synopsis of the paper. It is very vague in its present form.

The methodology of the proposed model must be illustrated by a clear flowchart.

Besides, the writing of the paper, including contributions, and methodologies, should be clearer and highlight the innovation of methods & principles.

Insufficient literature is presented to support the aim of the study. This point still needs further revision.

This paper doesn't detail the data collection, preparation, processing, and modeling procedure. Was the data collected using simulation or experimentation? The experimental setup, photo evidence, and procedure should be included.

Was the data normalized/ standardized?

It is unclear how much data was used for training and testing. What was the split? Additionally, results must be provided considering different holdout % and holdout validation approaches. Refer to this article to understand the holdout validation approach. You may refer to the following article https://doi.org/10.1115/1.4051696

Hyperparameters of the designed network must be included in a tabular form. You may refer to the paper https://www.techscience.com/CMES/v136n1/51215

Was the algorithm trained using standard hyperparameters, or were they altered?

Comment on computational time and complexity in the training of the algorithm.

The manuscript is more like a report than a research paper failing in solid discussion. Revise results and discussion part by critically examining results and including inferences drawn. 

Reviewer 2 Report

In this work, Li et al. employed deep learning methods to realize the inverse design of micro phononic beam relied on its size effects. This work proposed an interesting idea of training a NN to guide the choice of design parameters to achieve desired properties, e.g. band gap. Despite the idea of great interest to the audience, a few comments/confusions need response:

1.     Check spellings and wording:
A few expressions and potential typos are observed, e.g. “various frameworks of NNs are proposed by combing the MLP and CNN…”. Do the authors mean “combining the MLP and CNN”? Similar issues should be checked throughout this paper

2.     Design of the experiment & desired result:
As the authors indicated, the Forward path NNs will try to learn the functionality of Equation (12), then allow another NN to use the desired property (e.g. band gap) as input, and generate design parameters to produce such property. The quality of NN is purely based on the quality of dataset, and the optimization process. It’s not hard to imagine that the desired properties have no corresponding data in the training set to be fed into Forward Path. How would the authors guarantee a feasible design parameters to be produced from the Inverse Path?

3.     Training:
If not confused, there are 12^3 = 1728 data points used for training. How would the authors ensure the generalizability of these NNs? Or actually there is larger dataset?

Reviewer 3 Report

This manuscript aims to describe new research on algorithms for finding and developing novel photonic crystals (PnC) and metamaterials (MM). While there is no problem with English language, the content of the manuscript is not so clearly written, as the following concerns arise. 

The abstract promises to develop new material compounds for vibration and sound reduction, however, the conclusions does not give any hints how this could be realized. Please provide specific examples, how the method described in this manuscript contribute to this topic.

A fundamental information is missing on this manuscript on numerical networks. It has slightly improved in the revised version, but it is still unclear: What material data were used as the training data, what were used as the test data? 

Line 255 Please specify the name of the material.

Line 257: There are four design parameters introduced in table 1, but when the four examples are considered, they are not mentioned any more

Line 258: Please provide formulas for the ratios, or explain.

Line 264: Please specify the name of the materials in table 2

Figure 11 to 14: Concerning the material combinations, the figures all look very similar. The description of figures 11 to 14 is lacking in explanation, especially concerning the question, which the criterion to select a good material combination is

Especially: Why are the geometrical parameters defined in table 1 not mentioned for analysis of these figure?

Lines 486 to 490 are too vague and unspecific. Please explain, what the criteria for vibration and sound reduction are, and how this manuscript could contribute for this research.

Line 534 and so forth: Reference list

There are many more essential papers published on the inverse design of topology of PnC and MM, which should be cited as well, for example:

Muhammad et al. J. Phys. D: Appl. Phys. 55 (2022) 015106, 10.1088/1361-6463/ac9ce8

Fushan Lu,et al., J. Physics Conf. Ser., 2384 (2022) 012045, 10.1088/1742-6596/2384/1/012

Sunae So, Jungho Mun, Junsuk Rho, ACS Appl. Mat. Interfaces (2019) 11, 27, 24264, 10.1021/scsami.9b05857

On the other hand, the research group is  a very active group and has published many papers in the past. Why they are not cited?

Round 2

Reviewer 1 Report

All my comments have been addressed 

Author Response

Thanks for the reviewer's review.

Reviewer 3 Report

The manuscript has been improved compared to the previous version.